# Expanding the Spectrum of Autosomal Dominant *ATP6V1A*-Related Disease: Case Report and Literature Review

**DOI:** 10.3390/genes15091219

**Published:** 2024-09-18

**Authors:** Fabio Sirchia, Ivan Taietti, Myriam Donesana, Francesco Bassanese, Andrea Martina Clemente, Eliana Barbato, Alessandro Orsini, Alessandro Ferretti, Gian Luigi Marseglia, Salvatore Savasta, Thomas Foiadelli

**Affiliations:** 1Department of Molecular Medicine, University of Pavia, 27100 Pavia, Italy; fabio.sirchia@unipv.it; 2Medical Genetics Unit, Fondazione IRCCS Policlinico San Matteo, 27100 Pavia, Italy; 3Pediatric Unit, Department of Clinical, Surgical, Diagnostic, and Pediatric Sciences, University of Pavia, 27100 Pavia, Italy; myriam.donesana@gmail.com (M.D.); francesco.bassanese01@gmail.com (F.B.); andreamartina.clemente@gmail.com (A.M.C.); eliana.barbato01@universitadipavia.it (E.B.); gianluigi.marseglia@unipv.it (G.L.M.); t.foiadelli@smatteo.pv.it (T.F.); 4Pediatric Clinic, Fondazione IRCCS Policlinico San Matteo, 27100 Pavia, Italy; 5Pediatric Clinic, Department of Clinical and Experimental Medicine, University of Pisa, 56100 Pisa, Italy; aorsini.md@gmail.com; 6Pediatric Sleep Disease Centre, Child Neurology, NESMOS Department, School of Medicine and Psychology, Sapienza University of Rome, S. Andrea Hospital, 00189 Rome, Italy; alessandro.ferretti@uniroma1.it; 7Pediatric Clinic and Rare Diseases, P.O. Pediatrico Microcitemico “A. Cao”, Università degli Studi di Cagliari, 09121 Cagliari, Italy; salvatore.savasta@unica.it

**Keywords:** *ATP6V1A* mutation, speech delay, intellectual disability

## Abstract

Background: Developmental and epileptic encephalopathies (DEE) are a group of disorders often linked to de novo mutations, including those in the *ATP6V1A* gene. These mutations, particularly dominant gain-of-function (GOF) variants, have been associated with a spectrum of phenotypes, ranging from severe DEE and infantile spasms to milder conditions like autism spectrum disorder and language delays. Methods: We aim to expand *ATP6V1A*-related disease spectrum by describing a six-year-old boy who presented with a febrile seizure, mild intellectual disability (ID), language delay, acquired microcephaly, and dysmorphic features. Results: Genetic analysis revealed a novel de novo heterozygous pathogenic variant (c.82G>A, p.Val28Met) in the *ATP6V1A* gene. He did not develop epilepsy, and neuroimaging remained normal over five years of follow-up. Although *ATP6V1A* mutations have traditionally been linked to severe neurodevelopmental disorders, often with early-onset epilepsy, they may exhibit milder, non-progressive phenotypes, challenging previous assumptions about the severity of *ATP6V1A*-related conditions. Conclusions: This case expands the known clinical spectrum, illustrating that not all patients with *ATP6V1A* mutations exhibit severe neurological impairment or epilepsy and underscoring the importance of including this gene in differential diagnoses for developmental delays, especially when febrile seizures or dysmorphic features are present. Broader genotype–phenotype correlations are essential for improving predictive accuracy and guiding clinical management, especially as more cases with mild presentations are identified.

## 1. Introduction

Developmental and epileptic encephalopathies (DEEs) are a clinically and genetically heterogeneous group of disorders in which seizures interfere with neurological and cognitive development. DEEs often arise from de novo mutations (e.g., *SCN1A*, *GLUT1*, *KNQ2*) with a broader phenotypic spectrum than initially believed, as demonstrated by various Next Generation Study (NGS) studies [1,2]. Nevertheless, DEE diagnosis relies on clinical and neurophysiological criteria [3]. In particular, seizure activity can impact cognitive functions, disrupt brain networking, especially in the hippocampus, and cause deficits in brain plasticity [4]. There are numerous causative genes, and most of the mutations involved are de novo. It has been widely demonstrated that in cases of non-consanguineous parents, most patients with genetic syndromes present de novo mutations. This is particularly true for developmental disorders, where nearly half of the affected patients have de novo mutations in dominant autosomal genes [5,6]. One of these genes is the *ATP6V1A* gene, whose dominant de novo heterozygous variants with gain-of-function (GOF) have been recently linked to DEEs and neurodegeneration [7,8]. Through the years, *ATP6V1A* gene mutations have been described in patients with other phenotypes, such as cutis laxa, arising from recessive de novo or autosomal recessive loss-of-function (LOF) mutations [9], or autism spectrum disorder [10] and cerebral visual impairment [11] arising from dominant de novo GOF mutations. Moreover, mild phenotypes have been reported in the literature. Thus, a genetic variant in *ATP6V1A* does not necessarily imply a DEE [3]. Guerrini et al. [12] described the biggest cohort of *ATP6V1A* gene variants patients, widening the acknowledgment of this disease from a clinical and genetic/molecular point of view.

To date, it is clear that there is a need to expand the *ATP6V1A*-related disease spectrum describing the specific clinical phenotype associated with novel mutations not previously reported.

## 2. Case Presentation

We report the case of a six-year-old boy who presented in January 2018 at the pediatric emergency department for a bilateral tonic-clonic febrile seizure during an Influenza A infection (Table 1). The seizure lasted for five minutes and resolved spontaneously. The post-critical phase was characterized by confusion, transient hypertonus with hyperreflexia, and complete amnesia. He was born at term by elective Cesarian section from consanguineous parents of Northern African origins after an uneventful pregnancy. Birth weight, length, and head circumference (36 cm [+1 Standard Deviation, SD]) were within the normal ranges. He attended kindergarten, and his psychomotor development was reported to be normal until five years of age, when he was diagnosed with an expressive and receptive language disorder and mild intellectual disability (ID) (total IQ = 64, verbal IQ = 55, performance IQ = 76). Two years before, the parents reported a transient episode of “absence” during a fever, with a temporary loss of consciousness lasting less than one minute that had not been further investigated.

On examination, he had microcephaly (head circumference 48 cm, −2.2 SD) and dysmorphic features (long face, mild malar hypoplasia, slightly down-slanting palpebral fissures, mild hypotelorism, ears lifted, and enamel hypoplasia) (Figure 1). On neurological examination, he presented signs of gross and fine motor dyspraxia. Cerebral spinal fluid (CSF) chemical, physical, and microbiological investigations were normal. Prolonged video-electroencephalography (EEG) recordings showed slow high-voltage theta background activity, with bilateral multifocal spikes prevalent on the left hemisphere (Figure 2). Brain magnetic resonance imaging (MRI) (Figure 3), audiometric testing, and complete abdomen ultrasound were normal. Single nucleotide polymorphism (SNP) arrays were negative for microdeletions or microduplications. NGS analysis of a multigene epilepsy panel detected a de novo heterozygous variant c.82G>A (p.Val28Met) in the *ATP6V1A* gene. This variant is absent from the Genome Aggregation Database (gnomAD) and is classified as pathogenic based on the American College of Medical Genetics and Genomics (ACMG) criteria (PVS1, PM2, PP2, PP3, PM6).

After five years of follow-up, the patient is seizure-free, still presents mild non-progressive ID and language delay, minimal dyspraxic notes and difficulties in executing the commands, did not develop any cerebral alterations at the MRI imaging of the brain, and never developed any other neuropsychiatric comorbidities such as Attention Deficit Hyperactivity Disorder (ADHD) or Oppositional Defiant Disorder (ODD).

## 3. Evidence from the Literature

### 3.1. Effects of ATP6V1A Mutations

The pathogenesis involves the effects of v-ATPase in lysosomal homeostasis and neuronal connectivity. V-type proton (H+) ATPase (V-ATPase) is an ATP-dependent H+ pump that establishes and maintains the acidic environment of intracellular organelles (including secretory granules, endosomes, and lysosomes) and extracellular compartments. It is a fundamental component of the synaptic vesicles, where it allows neurotransmitter loading and regulates synaptic transmission [7]. Pathogenic mutations might induce specific synaptic defects, resulting in aberrant neural connectivity and altered synaptic plasticity, possibly causing seizures and cognitive impairment [13]. A physiological role in the surveillance of synaptic integrity and plasticity is hypothesized for *ATP6V1A*, as its depletion affects neurite elongation, stabilization, and function of excitatory synapses and prevents synaptic rearrangement upon the induction of plasticity. These observations might explain the associated neurodevelopmental diseases [14].

Notably, in vivo studies on knockdown zebrafish revealed several abnormalities, including suppression of acid secretion from the skin, growth retardation, trunk deformation, and loss of internal Ca^2+^ and Na^+^, highlighting the potential critical role of H^+^-ATPase in embryonic acid secretion and ion balance [15].

Moreover, *ATP6V1A* mutations have also been associated with an autosomal recessive form of metabolic cutis laxa syndrome. The intra-lysosomal environment V-ATPase-mediated acidification is fundamental for normal vesicular trafficking and for the activation of the enzymes involved in the glycosylation required for the assemblage of the extra-cellular matrix (ECM). The disruption of these processes results in abnormal glycosylation of serum proteins, intracellular accumulation of tropoelastin, reduced deposition of mature elastin in the ECM, accumulation of abnormal lysosomes and multivesicular bodies, and increased autophagy [9,16].

*ATP6V1A* is also a critical gene related to autophagy, which can induce autophagy through the activation of the mTOR signaling pathway [17]. Moreover, *ATP6V1A* is also reported to be involved in iron metabolism [18].

### 3.2. Clinical Phenotypes

De novo heterozygous *ATP6V1A* mutations have been recently associated with DEE, infantile spasms, autism, and childhood focal epilepsy with favorable outcomes [7,8,12,19,20].

To date, only a few patients with de novo *ATP6V1A* mutations have been described, with particular regard to Guerrini et al. [12], who reported 26 cases with de novo *ATP6V1A* mutations. Among them, 81% exhibited epilepsy (mostly early-onset), with fever-induced seizures as initial manifestations in 40%, while only one patient with fever-induced seizures onset did not develop epilepsy.

Febrile seizures seem to be a quite typical epileptic feature in these patients. Patients with epilepsy, in addition, often manifested mild to moderate developmental delay. DEEs are the main clinical evolution in patients with epilepsy (76%), with infantile spasms as the most common prominent seizures (85%). Moreover, severe developmental delay characterized most patients with DEE (63%) [12].

Language impairment seems to be a prominent feature of this genetic condition. Of note, not only the patients with epilepsy and DEE developed profound developmental delay and/or non-verbal status but also patients with mild phenotypes, namely febrile seizures or no seizures [12].

Moreover, acquired microcephaly and enamel dysplasia are frequently reported in *ATP6V1A*-related disorders (35% and 38%, respectively) [12].

Xiaoquan et al. reported six related patients with epilepsy without any other neurodevelopmental abnormalities. They were also proven to have good control with levetiracetam, potentially being crucial for the development of such patients [20]. Also, Li et al. described three new monoallelic *ATP6V1A* variants in people with childhood-onset focal epilepsy with good treatment response and favorable outcomes [21].

### 3.3. Neuroimaging

The most common MRI findings in these patients are hypomyelination and mild to severe local/diffuse encephalic hypoplasia/atrophy [7,12].

Only a few patients described had a normal brain MRI [8,12,20,21]. The few patients reported with normal brain MRI in most of the cases developed epilepsy and/or psychiatric disorders (e.g., ADHD, obsessive-compulsive disorder). Focusing on those patients, Kadwa et al. [8] reported a 7-month-old boy who presented with flexor spasms with loss of the previously acquired milestones, acquired microcephaly, no dysmorphism EEG suggestive of hypsarrhythmia with multifocal interictal discharges. Within the cohort of Guerrini et al. [12], five patients presented normal findings at brain MRI. The main clinical manifestations were mild–moderate ID, mild to severe epilepsy, enamel dysplasia, poor language/non-verbal status, and neuropsychiatric disorders. Of note, two of the patients presented with severe hypotonia and died prematurely. Conversely, the patients who presented with febrile seizures or who did not manifest seizures showed brain MRI abnormalities. Xiaoquan et al. [20] described a 9-month-old boy with non-drug-resistant epilepsy that arose with febrile seizures. Li et al. [21] reported three young males with different types of childhood-onset epilepsy with good clinical pharmacological treatment response.

## 4. Conclusions

We report a novel pathogenic *ATP6V1A* variant in a patient with a mild neurological phenotype. Unexpectedly, our patient never developed epileptic encephalopathy. Still, he presented febrile seizures and developed only language delay and mild non-progressive ID without epilepsy nor abnormalities on brain MRI over the years. He also developed acquired microcephaly and enamel dysplasia, which are frequently reported in *ATP6V1A*-related disorders.

As *ATP6V1A*-related phenotypes are being better described, it is crucial to consider both sides of the spectrum and highlight that not only severe neonatal encephalopathy and DEE but also very mild non-progressive phenotypes are possible. As already experienced with other monogenic causes of DEE, the initial phenotype descriptions tend to be biased by the selection of more severe clinical presentations. However, a deeper knowledge of the phenotypic spectrum is essential for future genotype–phenotype correlations, as well as prognostic or reproductive counseling. Considering reproductive counseling, in our case, we advised the family to perform prenatal invasive testing for potential future pregnancies, as we cannot rule out germline mosaicism in the parents, thus leading to a recurrence risk of 1–2% [22]. Moreover, the eventual new *ATP6V1A* variants, particularly of unknown significance (VUS) or likely pathogenic, should be validated through functional studies, like RNA-sequencing, that could enable researchers to further elucidate the complex mechanisms of this genetic condition [23,24]. We strengthen the need to consider *ATP6V1A*-related diseases in the differential diagnosis of developmental delay without epilepsy (e.g., by including this gene in an NGS dedicated panel) when febrile seizures, acquired microcephaly, and/or enamel dysplasia are present.

## Figures and Tables

**Figure 1 genes-15-01219-f001:**
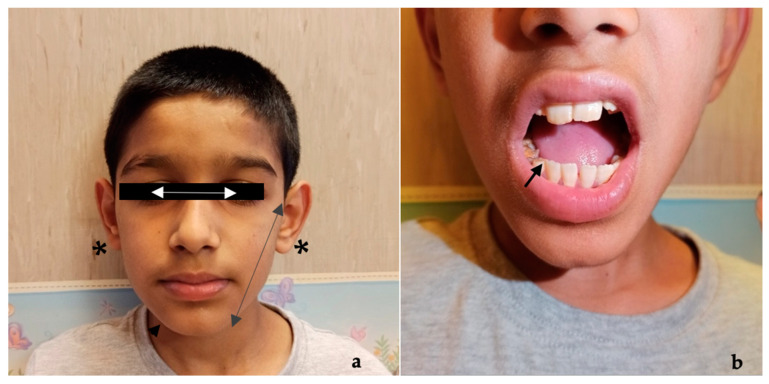
Clinical phenotype of the patient showing microcephaly, long face (
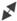
), mild malar hypoplasia (
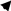
), mild hypotelorism (
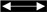
), and lifted ears (*****) (**a**), with particular focus on teeth characterized by enamel dysplasia (
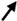
) (**b**).

**Figure 2 genes-15-01219-f002:**
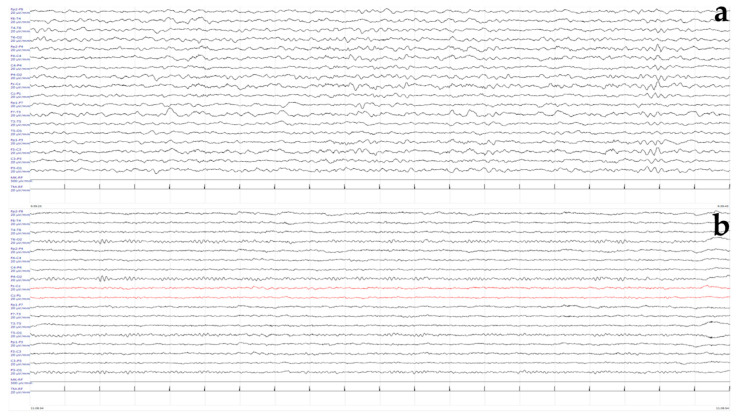
Standard 19-electrode EEG (20 s/pag—20 µV/mm). (**a**) first sleep EEG at the age of 5 years. Normal background sleep activity during N2 NREM phase, adequate representation of the physiological spindles, with some superimposed delta waves on the fronto-temporal regions. (**b**) The last follow-up wake EEG at the age of 11 years. Normal background activity with occipital alfa rhythm at 8 Hz. No sign of slow or epileptiform abnormality.

**Figure 3 genes-15-01219-f003:**
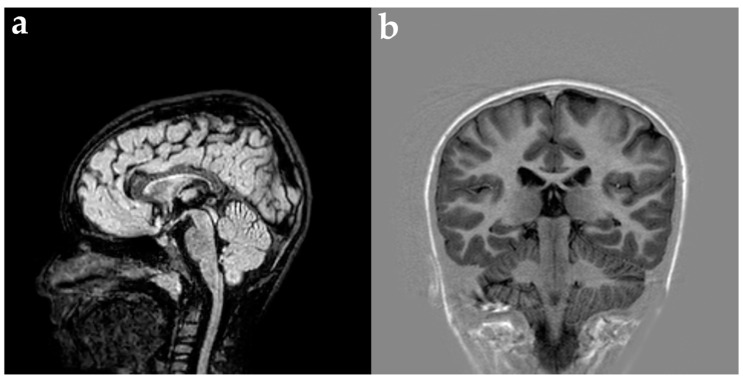
Magnetic Resonance Imaging (MRI) of the reported patient (**a**) sagittal Fluid Attenuated Inversion Recovery (FLAIR) sequence; (**b**) axial T1-weighted inversion recovery (IR) sequence. MRI showed no typical alterations of *ATP6V1A* patients (e.g., hypomyelination and encephalic hypoplasia/atrophy).

**Table 1 genes-15-01219-t001:** Summary of the clinical features of the patient.

Age (years), sex	8, male
De novo *ATP6V1A* mutations	c.82G>A (p.Val28Met)
Clinical diagnosis	Fever-induced seizures; expressive and receptive language disorder and mild intellectual disability.
Dysmorphic features	Microcephaly, long face, mild malar hypoplasia, slightly downloanting palpebral fissures, mild hypotelorism and ears lifted and enamel hypoplasia
Head circumference	At birth: 36 cm (+1 SD) At 8 years: 49 cm (−2.4 SD)
Age/symptoms at first clinical presentation	5 years, language delay, mild ID
Age at seizure onset	4 years
Seizures types	Bilateral tonic-clonic febrile seizures
Interictal EEG	Slow high voltage theta background activity with bilateral multifocal spikes, prevailing on the left hemisphere
Brain MRI	Normal at 6 years and 11 years
Clinical phenotype at last follow-up	Mild ID (DQ: 64), mild language delay, minimal dyspraxic notes and difficulties in executing the commands.

## Data Availability

The original contributions presented in the study are included in the article, and further inquiries can be directed to the corresponding author (ivan.taietti@gmail.com).

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
