# Peer review of "Expanding the Spectrum of Autosomal Dominant ATP6V1A-Related Disease: Case Report and Literature Review"

_genes, 2024, doi:10.3390/genes15091219_

Round 1

Reviewer 1 Report

Comments and Suggestions for Authors

This is a case report where the authors report the case of a patient who presented at the pediatric emergency department for a bilateral tonic-clonic febrile seizure during Influenza A infection. Genetic analysis revealed a novel de novo 27 heterozygous pathogenic variant (c.82G>A, p.Val28Met) in the ATP6V1A gene.

A couple of improvements that the authors might consider:

1. The authors never tried to do RNAseq from the blood. This can be very useful, like in this study (PMID: 23515576), and the authors should discuss RNAseq as an opportunity to analyse transcriptomic changes related to the disease.

2. De Novo variants are prevalent in medical genetics, as indicated in this paper (PMID: 30675999) for one Mendelian disease. The authors should also address the impact of de novo variants in clinical genetics.

Author Response

1. The authors never tried to do RNAseq from the blood. This can be very useful, like in this study (PMID: 23515576), and the authors should discuss RNAseq as an opportunity to analyse transcriptomic changes related to the disease.

We thank the reviewer for the comment. We did not perform RNA sequencing because the de novo heterozygous variant c.82G>A (p.Val28Met) in the ATP6V1A gene was determined to be pathogenic based on the American College of Medical Genetics and Genomics (ACMG) criteria (PVS1, PM2, PP2, PP3, PM6). However, conducting RNA sequencing from the blood could be valuable for further research, especially when dealing with patients who have VUS or likely pathogenic variants associated with potential ATP6V1A-related disorders. We address this issue in lines 214-217.

2. De Novo variants are prevalent in medical genetics, as indicated in this paper (PMID: 30675999) for one Mendelian disease. The authors should also address the impact of de novo variants in clinical genetics.

We thank the reviewer for the comment. We agree to the fact that de novo variants are crucial for understanding genetic disorders, particularly in cases where a family history of the disease is absent. We address this issue in lines 50-53.

Reviewer 2 Report

Comments and Suggestions for Authors

The current manuscript is a case report that describes a six year old child (male) with a heterozygous pathogenic variant (c.82G>A, p.Val28Met) in the ATP6V1A gene who does not have the severe epileptic phenotype that is typically associated with ATP6V1A gene mutations.

I have some comments that would strengthen the article.

Minor comment:

1, Avoid starting a sentence with an abbreviation i.e., line 119.

2, Line 41-43: Can the authors suggest how seizures may interfere with neurological and cognitive development in DEE?

3, Line 56 to 57: The authors should consider elaborating further on the ATP6V1A gene variants especially with regards to the genotype/phenotype correlations. Especially (I) are there any genotype-phenotype correlations with the ATP6V1A gene - i.e. are some pathogenic mutations expected to be more clinically severe based on a given mutational profile? If so, what is the basis for this i.e. are certain regions of the ATP6V1A gene more ‘vulnerable’ regarding the clinical phenotype? (II) Could the authors clarify whether a diagnosis of DEE (where there is suspected ATP6V1A involvement) can be based on diagnostic criteria rather than a genetic one?

4, Line 59 (Case Presentation): Did the patient present with any other neuropsychiatric comorbidity such as ADHD or ODD? If not, please make this clear.

5, Line 75: Figure 1: I note that the IRB statement indicates that the ethical review and approval were waived for this case report and that written consent for participation in this case report was obtained from the subject and his family (line 202-203). However, the manuscript should clearly state that the authors are allowed to use the participant’s photo (Figure 1a, b). Despite the eyes being covered, I feel that the case could still be identified and therefore obtaining the appropriate approval is mandatory. Please can the authors clarify and add the appropriate consent for use of Figures 1 and 2. 

6, Moreover, were the legal guardian/carer(s) involved in the development of the manuscript i.e. was it sent to them for review and their feedback incorporated? I would recommend that the authors elaborate on these points.

7, Line 94-96: The phenotype of the participant is subjective. For example, it is quite difficult to ascertain how the authors concluded that the patient had a long face, mild malar hypoplasia, mild hypotelorism, and lifted ears especially when some of the features in Figure 1 appear to be mild. Could the authors clarify this better? I think adding arrows in Figure 1 pointing to each of these milder features might help to better understand the case. Also, what is the typical clinical phenotype? It would be useful if the typical phenotype of ATP6V1A involvement could be compared to that seen here.

8, Line 124-127: This seems out of place here and I question whether describing studies done in zebrafish has any relevance to the case presented. I understand that these models can be useful but any inferences from such studies or other animal studies and how they relate to the patient phenotype should be treated with caution.

9, Section 4.2: It should be made clear how the current case presentation fits in with the existing clinical phenotypes. The authors state in their discussion that the case identifies a novel de novo heterozygous pathogenic variant in the ATP6V1A gene. This finding should be placed into context with other mutations. Are there any other cases with pathogenic variants similar to that identified (p.Val28Met) here that also had mild features, and unremarkable epilepsy and neuroimaging findings?

10, Section 4.3: The authors should bring in the findings from the case (normal neuroimaging) and compare to others in the literature especially for those patients with normal brain MRI but went on to develop epilepsy (line 170 – 171).

11, Line 189: Can the authors expand on what the ‘reproductive counselling’ could be given that the pathogenic variant was de novo?

Comments on the Quality of English Language

The manuscript would benefit from minor English language editing.

Author Response

1. Avoid starting a sentence with an abbreviation i.e., line 119.

We thank the reviewer for the comment. The correction will be found in line 126.

2. Line 41-43: Can the authors suggest how seizures may interfere with neurological and cognitive development in DEE?

We thank the reviewer for the comment. We discuss this issue in lines 45-47.

3. Line 56 to 57: The authors should consider elaborating further on the ATP6V1A gene variants especially with regards to the genotype/phenotype correlations. Especially (I) are there any genotype-phenotype correlations with the ATP6V1A gene - i.e. are some pathogenic mutations expected to be more clinically severe based on a given mutational profile? If so, what is the basis for this i.e. are certain regions of the ATP6V1A gene more ‘vulnerable’ regarding the clinical phenotype? (II) Could the authors clarify whether a diagnosis of DEE (where there is suspected ATP6V1A involvement) can be based on diagnostic criteria rather than a genetic one?

We thank the reviewer for the comment.

While it is easier to predict a loss of function of the protein and correlate it with the autosomal recessive form (cutis laxa) in the case of nonsense variants, it is much more difficult to predict the phenotype from possible gain of function or loss of function in missense mutations. Fassio et al. (2018) described a patient who had a mutation affecting the amino acid immediately preceding that of our patient (P27R) in the A subunit of the V1 domain. Molecular modeling predicted that the P27R variant would perturb subunit interaction. For this reason, it is likely a gain of function effect of the variant found in our patient but it becomes difficult to establish a clear genotype/phenotype correlation in the absence of functional tests, which we believe would be useful but are beyond the scope of our work.

Developmental and epileptic encephalopathy (DEE) refers to when both developmental impairment and epileptic activity have an impact on the cognitive and behavioral state of the affected person. Most of them recognize a genetic etiology, whereby the genetic variant is responsible for both cognitive impairment and severe epilepsy. As we described in the manuscript, ATP6V1A variants can lead to milder phenotypes without the development of DEE, such as in our case. Thus, DEE diagnosis must rely on clinical and neurophysiological criteria, as genetic variants of ATP6V1A do not necessarily correlate with DEE. We address this topic in lines 45-46 and 58.

4. Line 59 (Case Presentation): Did the patient present with any other neuropsychiatric comorbidity such as ADHD or ODD? If not, please make this clear.

We thank the reviewer for the comment. The patient never developed others neuropsychiatric comorbidity such as ADHD or ODD. That specification will be found in lines 95-97.

5. Line 75: Figure 1: I note that the IRB statement indicates that the ethical review and approval were waived for this case report and that written consent for participation in this case report was obtained from the subject and his family (line 202-203). However, the manuscript should clearly state that the authors are allowed to use the participant’s photo (Figure 1a, b). Despite the eyes being covered, I feel that the case could still be identified and therefore obtaining the appropriate approval is mandatory. Please can the authors clarify and add the appropriate consent for use of Figures 1 and 2.

We thank the reviewer for the comment. We have revised the manuscript to explicitly mention that specific consent for the publication of these photographs was obtained and we added the appropriate statement in lines 233-235. We provided a scan of related written consent attached to the email response.

6. Moreover, were the legal guardian/carer(s) involved in the development of the manuscript i.e. was it sent to them for review and their feedback incorporated? I would recommend that the authors elaborate on these points.

We thank the reviewer for the comment. We confirm that the legal guardian(s) of the subject were actively involved in the development of this manuscript. A draft of the manuscript was sent to them for their review, and their feedback was considered and incorporated into the final version. We added an appropriate statement in lines 230-233.

7. Line 94-96: The phenotype of the participant is subjective. For example, it is quite difficult to ascertain how the authors concluded that the patient had a long face, mild malar hypoplasia, mild hypotelorism, and lifted ears especially when some of the features in Figure 1 appear to be mild. Could the authors clarify this better? I think adding arrows in Figure 1 pointing to each of these milder features might help to better understand the case. Also, what is the typical clinical phenotype? It would be useful if the typical phenotype of ATP6V1A involvement could be compared to that seen here.

We thank the reviewer for the comment. You will find arrows pointing to each milder feature in Figure 1. There is not a typical clinical phenotype (i.e. typical facies). We described the typical facial and skull abnormalities in lines 165-166, reported by Guerrini et al. that described the largest cohort of patients with ATP6V1A-related disease to date.

8. Line 124-127: This seems out of place here and I question whether describing studies done in zebrafish has any relevance to the case presented. I understand that these models can be useful but any inferences from such studies or other animal studies and how they relate to the patient phenotype should be treated with caution.

We thank the reviewer for the comment. Zebrafish model was included to widen the knowledge to date about the functional mechanism of ATP6V1A, which is implicated in the patient's phenotype. These studies provide insight into the gene's function in a living organism and help illustrate the potential biological processes affected by the mutation observed in the patient. Although we agree that inferences from animal studies should be made with caution, we believe that presenting these findings helps build a theoretical framework to understand how the gene may function. Thus, we rewrote the sentence in lines 131-134.

9. Section 4.2: It should be made clear how the current case presentation fits in with the existing clinical phenotypes. The authors state in their discussion that the case identifies a novel de novo heterozygous pathogenic variant in the ATP6V1A gene. This finding should be placed into context with other mutations. Are there any other cases with pathogenic variants similar to that identified (p.Val28Met) here that also had mild features, and unremarkable epilepsy and neuroimaging findings?

We thank the reviewer for the comment. One of the aims of this work is precisely to present a milder case compared to those described so far in the literature. As has already occurred with other genes responsible for neurodevelopmental disorders with epileptic encephalopathy, sometimes the phenotypic spectrum can be so broad that it does not cause the epileptic phenotype but only the neuropsychiatric one.

10. Section 4.3: The authors should bring in the findings from the case (normal neuroimaging) and compare to others in the literature especially for those patients with normal brain MRI but went on to develop epilepsy (line 170 – 171).

We thank the reviewer for the comment. We address this issue in lines 178-189.

The literature yields highly varied results, making them difficult to generalize. Due to the rarity of the condition, the potential clinical-radiological correlations remain unclear. Our case aims to broaden the understanding of this condition and contribute fresh data for future studies, which may help to clarify these aspects.

11. Line 189: Can the authors expand on what the ‘reproductive counselling’ could be given that the pathogenic variant was de novo?

We thank the reviewer for the comment. We address this issue in lines 211-214.

Round 2

Reviewer 2 Report

Comments and Suggestions for Authors

Thank you for revising the manuscript. Please check that Figure 1 has the arrows. It is mentioned that arrows point to each milder feature, however, I could not see the arrows in Figure 1 of the revised manuscript. Accordingly, Figure 1 legend should also be updated to signify what the arrows mean.

Comments on the Quality of English Language

The manuscript would benefit from some English language editing.

Author Response

Dear reviewers,

We have made the necessary minor revisions to the English language.

We have enhanced Figure 1 by adding arrows and other symbols, which can be found in the figure legend. Please inform us if any further revisions are necessary.

Kind regards, 

Ivan Taietti, MD